**Data Availability Statement:** All relevant data are within the manuscript and its Supporting Information files.

# Prevalence of male circumcision in four culturally non-circumcising counties in western Kenya after 10 years of program implementation from 2008 to 2019

Elijah Odoyo-June[1]*, Stephanie Davis[2], Nandi Owuor[3], Catey Laube[4], Jonesmus Wambua[3], Paul Musingila[1], Peter W. Young[1], Appolonia Aoko[1], Kawango Agot[5], Rachael Joseph[1], Zebedee Mwandi[3], Vincent Ojiambo[6], Todd Lucas[2], Carlos Toledo[2], Ambrose Wanyonyi[7]

1 U.S. Centers for Disease Control and Prevention (CDC), Division of HIV & TB, Nairobi, Kenya, 2 U.S. Centers for Disease Control and Prevention (CDC), Division of HIV & TB, Atlanta, GA, United States of America, 3 Jhpiego, Nairobi, Kenya, 4 Jhpiego, Baltimore, Maryland, United States of America, 5 Impact Research and Development Organization (IRDO), Kisumu, Kenya, 6 USAID, Nairobi, Kenya, 7 National AIDS and STI Control Program (NASCOP), Ministry of Health, Nairobi, Kenya

* yed0@cdc.gov

## Abstract

### Introduction

Kenya started implementing voluntary medical male circumcision (VMMC) for HIV prevention in 2008 and adopted the use of decision makers program planning tool version 2 (DMPPT2) in 2016, to model the impact of circumcisions performed annually on the population prevalence of male circumcision (MC) in the subsequent years. Results of initial DMPPT2 modeling included implausible MC prevalence estimates, of up to 100%, for age bands whose sustained high uptake of VMMC pointed to unmet needs. Therefore, we conducted a cross-sectional survey among adolescents and men aged 10–29 years to determine the population level MC prevalence, guide target setting for achieving the goal of 80% MC prevalence and for validating DMPPT2 modelled estimates.

### Methods

Beginning July to September 2019, a total of 3,569 adolescents and men aged 10–29 years from households in Siaya, Kisumu, Homa Bay and Migori Counties were interviewed and examined to establish the proportion already circumcised medically or non-medically. We measured agreement between self-reported and physically verified circumcision status and computed circumcision prevalence by age band and County. All statistical were test done at 5% level of significance.

### Results

The observed MC prevalence for 15-29-year-old men was above 75% in all four counties; Homa Bay 75.6% (95% CI [69.0–81.2]), Kisumu 77.9% (95% CI [73.1–82.1]), Siaya 80.3%

**Funding:** This survey was funded by the President's Emergency Plan for AIDS Relief (PEPFAR) through the U.S. Centers for Disease Control and Prevention (CDC) under the terms of Cooperative Agreement # GH001469. "The author (s) received no specific funding for this work. The funders had no role in study design, data collection and analysis, decision to publish, or preparation of the manuscript.

**Competing interests:** The authors have declared that no competing interests exist.

(95% CI [73.7–85.5]), and Migori 85.3% (95% CI [75.3–91.7]) but were 0.9–12.4% lower than DMPPT2-modelled estimates. For young adolescents 10–14 years, the observed prevalence ranged from 55.3% (95% CI [40.2–69.5]) in Migori to 74.9% (95% CI [68.8–80.2]) in Siaya and were 25.1–32.9% lower than DMMPT 2 estimates. Nearly all respondents (95.5%) consented to physical verification of their circumcision status with an agreement rate of 99.2% between self-reported and physically verified MC status (kappa agreement p-value<0.0001).

## Conclusion

This survey revealed overestimation of MC prevalence from DMPPT2-model compared to the observed population MC prevalence and provided new reference data for setting realistic program targets and re-calibrating inputs into DMPPT2. Periodic population-based MC prevalence surveys, especially for established programs, can help reconcile inconsistencies between VMMC program uptake data and modeled MC prevalence estimates which are based on the number of procedures reported in the program annually.

## 1. Introduction

Voluntary medical male circumcision (VMMC) reduces sexual transmission of HIV from women to men by approximately 60% [1–3]. VMMC is currently implemented as a component of HIV prevention programs in 15 sub Saharan African countries with high HIV prevalence and low historical rates of male circumcision (MC). Since 2008, Kenya has prioritized VMMC services primarily for four counties in western Kenya that are mainly inhabited by the Luo ethnic group that do not practice circumcision culturally and have high HIV prevalence (13.0–19.6%) [4]. Additionally, VMMC is implemented in focal areas of the Rift Valley region and Nairobi where migrant or indigenous non-circumcising populations live. Kenya's VMMC program achieved 92% of its service delivery target for the first 5-year national strategic plan (2008–2013) and met its annual targets in the second strategic plan (2014–2019) thereby reporting over two million cumulative circumcisions as of September 2018 [5].

Despite excellent performance against annual program targets, lack of accurate population level MC prevalence data by key age bands continued to hamper VMMC program planning and impact assessment in Kenya. Therefore, in 2016 the VMMC decision makers program planning tool version 2 (DMPPT2) was used to model the impact of cumulative circumcisions performed on the prevalence of MC and to guide subsequent annual target setting in four top VMMC priority counties in western Kenya (namely, Siaya, Kisumu, Homa Bay and Migori). DMPPT2 is a compartmental mathematical model that uses the number of men circumcised annually by 5-year age bands, adjusted for age progression, and mortality plus migration, to estimate changes in MC prevalence [6, 7] and models the impact of additional VMMCs on the MC prevalence in a given geographic area. The first DMPPT2 modelling in Kenya conducted in 2016 and published in 2018 [8] generated unexpected high MC prevalence estimates up to and exceeding 100% for some age some age groups in the four counties. Moreover, contrary to the expectation that uptake of VMMC would decline as MC prevalence approached 100%, stable uptake of VMMC services was observed in age bands and geographic area that had achieved or nearly achieved 100% MC prevalence according to DMPPT2. Divergence between sustained high VMMC uptake data and high MC prevalence from DMPPT2 model persisted despite adjustment for known potential confounders. For example, MC prevalence estimates

in Homa Bay county remained higher than expected even after adjustment for in-migration from surrounding areas to access VMMC services, duplicate reporting, errors in population estimates and replacement of traditional circumcision with medical circumcisions in the VMMC program. These data issues were first identified in Kenya because of its high baseline circumcision prevalence and availability of reliable data which made it possible to adjust for migration and replacement of traditional circumcision with VMMC in the coverage estimates. The use of DMPPT2 in other African countries, including South Africa and Mozambique, also revealed instances of high MC prevalence above 80%, but the insights on inconsistency between VMMC program uptake and DMPPT2 modeled MC prevalence were revealed by triangulation and a granular analysis of program data in Kenya.

Like immunization coverage estimation [9], VMMC coverage estimation requires reconciliation of service delivery data on the number of procedures performed with population survey data to determine if the number of procedures reported match well with the changes in population coverage.

In order to resolve the discrepancy between sustained VMMC uptake data and high MC prevalence estimates from DMPPT2 model, we conducted a population-based survey to get reference MC prevalence data by age bands for setting realistic program targets and validating DMPPT2 inputs.

As a secondary objective, we sought to assess the accuracy of self-reported circumcision as the primary data source for determining MC prevalence. Although data from previous surveys in Kenya suggest that self-reported circumcision status is generally reliable, those studies were limited by high participant non-response rate [10, 11]. Thus, we also sought to assess the reliability of self-reported circumcision status using physical verification as a reference. The specific objectives of the survey were to 1) Estimate prevalence of self-reported circumcision among adolescents and men 10–14 and 15–29 years of age, 2) Assess the accuracy of self-reported MC status using physically verified circumcision status, and 3) Evaluate the association between circumcision status and demographic characteristics of adolescents and men 10–29 years.

## 2. Methods

From July to September 2019 we conducted a cross-sectional household survey among adolescents and men aged 10–29 years in Siaya, Kisumu, Homa Bay and Migori counties in Kenya. We used a structured questionnaire to collect data on demographic characteristics, knowledge of VMMC, service delivery experience of circumcised men and outcome of circumcision status verification. Questionnaire contents were developed under the leadership of the national VMMC technical working group with technical input from various stakeholders including VMMC service providers and researchers. Reported circumcisions were coded as medical if conducted by a health worker otherwise non-medical. The final questionnaire was translated from English into Kiswahili and Luo language and back translated into English to ensure accuracy. The questionnaires were further refined after field pretesting then distributed along with consent forms for use by the trained research assistants (S2 File).

A two-stage cluster sampling approach was used with enumeration areas (EAs) as the primary sampling units from which households were selected. EAs are small counting units of 50–149 households cartographically mapped by the Kenya National Bureau of Statistics (KNBS) to facilitate the 2009 Kenya Population Housing Census [11]. Within each county, the EAs are divided into rural and urban strata as defined in the KNBS Fifth National Sample Survey, and Evaluation Program (NASSEP V) 2009 Kenya Population and Housing Census master sampling frame [12]. In the first stage, a total of 77 rural and 46 urban EAs (clusters) were

randomly selected from the four target counties using probability proportional to size of total population sampling methodology in the 2009 Kenya population housing census. This ensured that the survey was designed to produce representative estimates for MC prevalence at county level [13].

A team composed of KNBS staff, a community health volunteer, a village health committee member, and a trained research assistant (RA) visited all sampled EAs, listed all households (new and old) for number of males and females to help update KNBS NASSEP IV, assigned a unique identifier, and their Global Positioning System (GPS) coordinates obtained. The collected household GPS coordinates were used during the survey to relocate sampled households for survey data collection. After updating the households which form the new sampling frame in the selected EA for the second sampling stage, systematic random sampling was then used to select 48–50 households from each EA from the new household listing, thus deduplication was unnecessary. Using findings from the 2009 census indicating that the average household size was four, it was projected that on average, there would be at least one man aged 15–29 years residing in each household, and one younger adolescent boy aged 10–14 years residing in every other household. In order to oversample young adolescents to achieve comparable precisions of estimates between the two target age groups, enrollment was offered to all younger adolescents aged 10–14 years residing in all selected households, and to men aged 15–29 years residing in every other selected household. A resident was defined as a person who lived in the selected household as indicated by the head of the household, or who spent the previous night in the selected household. The head of household was defined as a usual resident member of the household, the key decision maker for the household and the person whose authority was acknowledged by all members of the household.

Trained RAs visited the selected households accompanied by community health volunteers and village health committee members who served as local guides. Households whose eligible occupants were unavailable at the time of first visit were scheduled for up to three follow-up visits on different days. Age eligibility was based on a self-report of 10–29 years. Households found to have no eligible participants were not replaced. Prior to enrollment, resident men aged 18–29 years provided written informed consent, and those 17 years or younger provided written assent in addition to written informed parental or guardian consent. Individuals who were unwilling or unable to provide informed consent or assent, and those with cognitive or hearing disabilities that would undermine participation in the survey were excluded. After enrollment, RAs administered the study questionnaire using password protected tablets installed with Research Electronic Data Capture (REDCap) software version 8.9 [14] to collect all data electronically. The questionnaire design into REDCap employed logic checks to increase data precision and consistency during data collection.

## 2.1 Verification of circumcision status

After completing the questionnaire, respondents were requested to give a written personal or parental consent plus assent, where applicable, for physical verification of circumcision status. To minimize the possibility of psychological harm or embarrassment from genital exposure and examination, the RAs explained the process to the participants before inviting them to give voluntary informed consent. For respondents who consented to MC status verification, a trained RA examined the penis using a standard job aid to classify circumcision status as: i) fully circumcised if the foreskin was completely absent leaving the glans completely uncovered, ii) partially circumcised if the amount of foreskin present partially covered the glans, or iii) uncircumcised if the foreskin covered the glans fully. For minors 17 years or younger, verification was conducted in the presence or absence of the parent or guardian depending on

participant or guardian/parent preference. As part of their training before the survey, RAs practiced physical assessment and classification of MC status under the guidance of a medical officer experienced in VMMC services provision. Photo illustrations of the external male genital anatomy and different sizes of foreskins were presented to the RAs before small group practice sessions on classifying foreskin as fully circumcised, partially circumcised, or uncircumcised. Finally, each RA was given a laminated job aid with photo illustrations of different grades of circumcision as a pocket guide to be used for reference during physical examination of respondents in the field. Uncircumcised participants were offered a flyer with information on the health benefits of VMMC and where to access VMMC services.

## 2.2 Data management

Collected data was reviewed routinely in the field, by the team lead and centrally by the study data manager for completeness, accuracy, and consistency. Data were imported into Stata version 15 [15] from REDCap for analysis. All data were checked for consistency and multiple imputations done for missing age records.

## 2.3 Data weighting

Weights were computed and applied during data analysis to adjust for household and individual-level non-response, and accounting for differences in probability of household selection. The design weights incorporated the probabilities of selection of the EAs from the updated 2009 census database, and the probabilities of selection of the households from each of the selected EAs.

The survey cluster weight was calculated using the updated EA selection probabilities for the i-th EA per stratum and also accounting for non-selection, the household weights were calculated using the updated household listing per EA in each stratum accounting for non-selection and lastly individual weights were calculated using the updated household listing for the adolescents aged 10–14 years and men 15–29 years by stratum while adjusting for non-participation. The overall cluster weight was obtained by multiplying the three obtained values at cluster, household and individual levels accounting for design effect. County specific sampling weights were calculated as inverse of the probability of selection of individuals in the EAs including response probability. Selection probabilities were calculated separately for each sampling stage and for each unit of sampling. Survey weights were computed separately for the interview and MC status verification at the county level. The survey final weights were normalized so that the total final weights equal to the total sample size. County population MC prevalence (both verified and self-reported) was calculated. Multiplying the value of each participant's survey response by the corresponding nonresponse-adjusted weight, then summing up the products across all units (clusters) and finally dividing by the sum of all weights per county. Therefore, we did not employ multiple comparison of the MC prevalence across the counties. However, each county MC prevalence was obtained accounting for survey design.

## 2.4 Data analysis

The MC status verification rates were computed by dividing the weighted verified MC rates with self-reported rates. Agreement between physically verified and self-reported MC status was calculated for individuals who participated in both the survey and MC verification using weighted Kappa statistics. Univariate and multivariable survey logistic regression were used to assess demographic and social factors associated with circumcision. All analyses were weighted and adjusted for the complex survey design to account for both stratification and clustering.

Age and county-specific MC prevalence are reported as point estimates with 95% confidence intervals. All statistical analysis tests were conducted at 5% level of significance.

## 2.5 Ethical considerations

Ethical approval for this survey was granted by the Maseno University Ethical Review Committee (MUERC). The survey protocol was also reviewed in accordance with the US centers for disease control and prevention (CDC) human research protection procedures and determined to be research. However, CDC investigators did not interact with human subjects or have access to identifiable data or specimens for research purposes.

## 3. Results

### 3.1 Sampling and response rate

Overall, 86.8%, (3,569 of 4,113) eligible adolescents and men participated in the survey. Response rate varied from 84.0% to 92.2% across the four counties. All respondents self-reported their circumcision status, and 3,410 (95.5%) consented to physical verification of their circumcision status. Table 1 shows the multistage sampling cascade, participant enrolment and response rate by county. Note: Numbers reported in this table are unweighted.

### 3.2 Socio demographic characteristics of respondents

Demographic and social characteristic of respondents by county are shown in Table 2. The overall median age of respondents was 14 years (interquartile range = 12–18), 89.6% were of Luo ethnicity; 76.6% had completed primary school; 16.7% secondary school and 6.7% post-secondary education. The majority (93.0%) had never married, 6.9% were married and 0.1% were divorced, separated, or widowed. A minority (13.5%) reported being employed. Under

**Table 1. Sampling cascade and response rate in the 2019 male circumcision prevalence survey in four counties in western Kenya.**

| Variable | County | | | | |
|---|---|---|---|---|---|
| | **Siaya** | **Kisumu** | **Homa Bay** | **Migori** | **Total** |
| **Total EAs by County** | 1,905 | 2,003 | 2,002 | 1,642 | 7,552 |
| Sampled EA's | 31 | 31 | 31 | 30 | 123 |
| Listed HH in all sampled EAs | 5,212 | 7,631 | 2,995 | 3,618 | 19,456 |
| Total sampled HH | 1,527 | 1,548 | 1,498 | 1,455 | 6,028 |
| Sampled HH with eligible men/men 10–29 years | 551 | 570 | 685 | 605 | 2,411 |
| Sampled HH with eligible boys/men 10–14 years | 398 | 357 | 505 | 474 | 1,734 |
| Sampled HH with eligible men 15–29 years | 268 | 342 | 369 | 342 | 1,321 |
| **Total number of eligible men 10–29 years** | **892** | **907** | **1,205** | **1,109** | **4,113** |
| Number of eligible boys 10–14 years | 542 | 439 | 675 | 635 | 2,291 |
| Number of eligible men 15–29 years | 350 | 468 | 530 | 474 | 1,822 |
| | N (%) | N (%) | N (%) | N (%) | N (%) |
| **Total eligible men interviewed 10–29 years** | **822 (92.2)** | **792 (87.3)** | **1,023 (84.9)** | **932 (84.0)** | **3,569 (86.8)** |
| Eligible boys 10–14 years interviewed | 514 (94.8) | 409 (93.2) | 606 (89.8) | 566 (89.1) | 2095 (91.4) |
| Eligible men15–29 years interviewed | 308 (88.0) | 383 (81.8) | 417 (78.7) | 366 (77.2) | 1474 (80.9) |
| **Total with verified MC status 10–29 years** | **806 (98.1%)** | **717 (90.5%)** | **983 (96.1%)** | **904 (97.0%)** | **3,410 (95.5%)** |
| Boys 10–14 years with verified MC status | 511 | 392 | 597 | 563 | 2,063 |
| Men 15–29 years with verified MC status | 295 | 325 | 386 | 341 | 1,347 |
| MC status verification rate 10–29 years | 98.1% | 90.5% | 96.1% | 97.0% | 95.5% |

**Table 2. Demographic characteristics of respondents in a male circumcision survey among 10-29-year-old boys and men from four counties, western Kenya, 2019.**

| Characteristic | Counties | | | | |
|---|---|---|---|---|---|
| | Homa Bay | Kisumu | Migori | Siaya | All Counties |
| | N (%) | N (%) | N (%) | N (%) | N (%) |
| **Age (years)** | | | | | |
| Median (Interquartile Range) | 14 (12–18) | 14 (12–20) | 14 (12–18) | 13 (12–18) | 14 (12–18) |
| 10–14 | 606 (59.2) | 409 (51.6) | 566 (60.7) | 514 (62.5) | 2095 (58.7) |
| 15–19 | 226 (22.1) | 175 (22.1) | 207 (22.2) | 158 (19.2) | 766 (21.5) |
| 20–24 | 125 (12.2) | 124 (15.7) | 99 (10.6) | 84 (10.2) | 432 (12.1) |
| 25–29 | 66 (6.5) | 84 (10.6) | 60 (6.4) | 66 (8) | 276 (7.7) |
| Total | 1023 (100) | 792 (100) | 932 (100) | 822 (100) | 3569 (100) |
| **Marital status*** | | | | | |
| Never married | 959 (94) | 721 (91.5) | 866 (93.2) | 727 (92.7) | 3273 (93.0) |
| Married | 60 (5.9) | 66 (8.4) | 63 (6.8) | 56 (7.1) | 245 (7) |
| Divorced, separated or Widowed | 1 (0.1) | 1 (0.1) | 0 (0) | 1 (0.1) | 3 (0.1) |
| Total | 1020 (100) | 788 (100) | 929 (100) | 784 (100) | 3521 (100) |
| **Highest level of education** | | | | | |
| Primary and below | 771 (75.4) | 571 (72.1) | 732 (78.5) | 659 (80.2) | 2733 (76.6) |
| Secondary | 173 (16.9) | 150 (18.9) | 159 (17.1) | 114 (13.9) | 596 (16.7) |
| Post-Secondary | 79 (7.7) | 71 (9) | 41 (4.4) | 49 (6) | 240 (6.7) |
| Total | 1023 (100) | 792 (100) | 932 (100) | 822 (100) | 3569 (100) |
| **Religion** | | | | | |
| Christian | 1001 (97.8) | 776 (98) | 924 (99.1) | 811 (98.7) | 3512 (98.4) |
| Other | 22 (2.2) | 16 (2) | 8 (0.9) | 11 (1.3) | 57 (1.6) |
| Total | 1023 (100) | 792 (100) | 932 (100) | 822 (100) | 3569 (100) |
| **Employment** | | | | | |
| Employed | 141 (13.8) | 135 (17) | 109 (11.7) | 96 (11.7) | 481 (13.5) |
| Not employed | 882 (86.2) | 657 (83) | 823 (88.3) | 726 (88.3) | 3088 (86.5) |
| Total | 1023 (100) | 792 (100) | 932 (100) | 822 (100) | 3569 (100) |
| **Ethnic group** | | | | | |
| Luo | 1001 (97.8) | 725 (91.5) | 685 (73.5) | 787 (95.7) | 3198 (89.6) |
| Non-Luo | 22 (2.2) | 67 (8.5) | 247 (26.5) | 35 (4.3) | 371 (10.4) |
| Total | 1023 (100) | 792 (100) | 932 (100) | 822 (100) | 3569 (100) |

Marital status* 48 participants with no recorded response were excluded; 3 in Homa bay, 4 in Kisumu, 3 in Migori and 38 in Siaya.

### 3.3 MC prevalence by county

Fig 1 shows the observed MC prevalence by county and two age bands, 10–14 and 15–29 years. The observed MC prevalence for 15-29-year-old men was above 75% in all four counties; Homa Bay 75.6% (95% CI [69.0–81.2]), Kisumu 77.9% (95% CI [73.1–82.1]), Siaya 80.3% (95% CI [73.7–85.5]), and Migori 85.3% (95% CI [75.3–91.7]). For 10–14-year-old boys, the observed prevalence ranged from 55.3% (95% CI [40.2–69.5]) in Migori to 74.9% (95% CI [68.8–80.2]) in Siaya County.

For all age bands, the observed MC prevalence results from this 2019 survey were lower than DMPPT2 modelled estimates for the same year and for 2016 [8]. Among 15–29 year old men, the population survey results were 0.9–12.4% lower than DMPPT2 estimates; 75.6 vs 76.5% in Homabay, 77.7 vs 100% in Kisumu, 90.3 vs 80.3% in Siaya and 85.3 vs 91.2% in

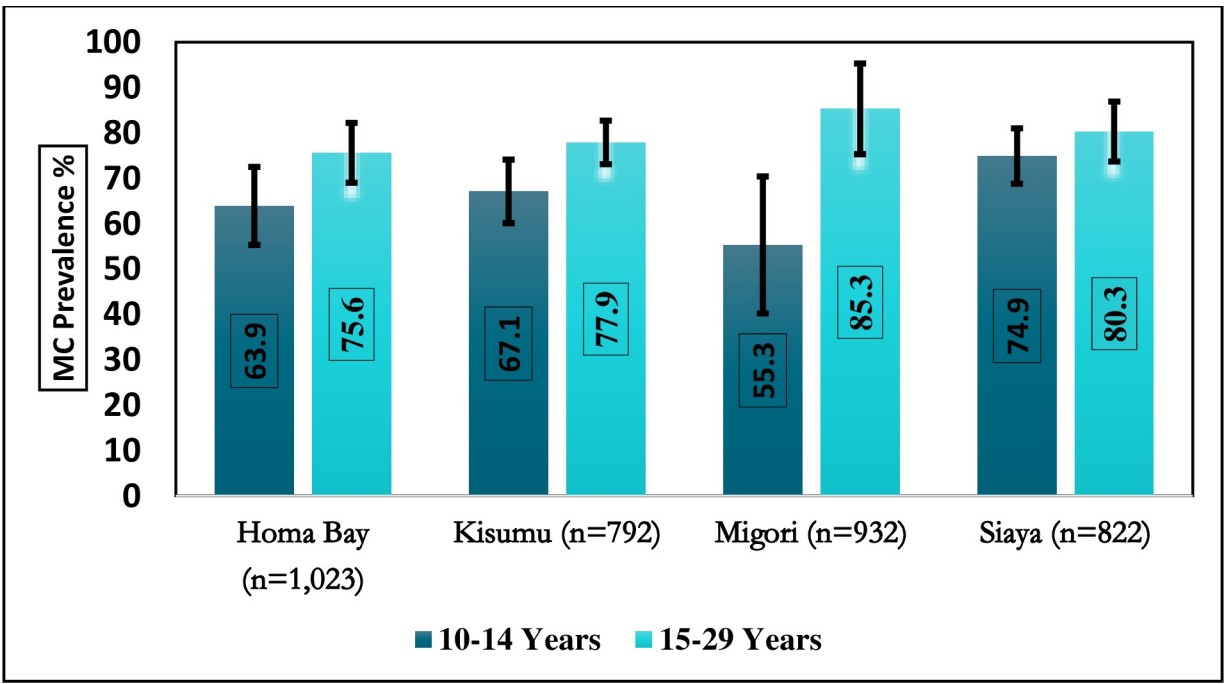

**Fig 1. Observed prevalence of MC by age group and county in western Kenya, 2019.**

Migori. For 10–14 year old boys, the prevalence results from the survey ranged from 55.3% (95% CI [40.2–69.5]) in Migori to 74.9% (95% CI [68.8–80.2]) in Siaya County and were lower than the 2019 DMMPT 2 estimate by 25.1–32.9% across all counties. The differences in 2019 MC prevalence estimates from the population survey and DMPPT2 model are presented in Table 3.

Granular details including the prevalence of MC by socio demographics characteristics of the respondents are presented in Table 4.

### 3.4 Contribution of non-medical circumcision to observed MC prevalence

The contribution of non-medical circumcision to the total circumcisions reported by county for the 10-14-year age band ranged from 2.1% in Siaya to 6.9% in Migori but was higher for the 15-29-year age band ranging from 3.1% in Homa bay to 28.4% in Migori (Table 5).

### 3.5 Agreement between self-reported and physically verified MC status

Overall, 95.5% (3,410/3,569) of the respondents consented to and were examined for physical verification of their circumcision status after completing the structured questionnaire. The MC status verification cascade is shown in Table 6. There was no difference in the proportion

**Table 3. Comparing MC prevalence data from 2019 population survey with DMPPT2 modelled estimates the same year.**

| | Siaya | | Kisumu | | Homabay | | Migori | |
|---|---|---|---|---|---|---|---|---|
| Age | MC Prev Survey | DMPPT2 estimate | MC Prev Survey | DMPPT2 estimate | MC Prev Survey | DMPPT2 estimate | MC Prev Survey | DMPPT2 estimate |
| **10–14 yrs** | 74.9 | 100.0 | 67.1 | 100.0 | 63.9 | 89.0 | 55.3 | 96.4 |
| **15–29 yrs** | 80.3 | 90.3 | 77.9 | 100.0 | 75.6 | 76.5 | 85.3 | 91.2 |

**Table 4.** MC prevalence among boy and men aged 10–29 years across selected demographics characteristics by county.

| Characteristic | Homa Bay Unweighted n/N | Homa Bay Weighted Prev. % (95% CI) | Kisumu Unweighted n/N | Kisumu Weighted Prev. % (95% CI) | Migori Unweighted n/N | Migori Weighted Prev. % | Siaya Unweighted n/N | Siaya Weighted Prev. % | Total Unweighted n/N | Total Weighted Prev. % |
|---|---|---|---|---|---|---|---|---|---|---|
| **Age** | | | | | | | | | | |
| 10–14 | 393/606 | 63.9(55.8–72.1) | 271/409 | 67.1(60.7–73.5) | 298/566 | 55.3(40.5–70.1) | 396/514 | 74.9(68.5–81.3) | 1358/2095 | 64.3(59.3–69.3) |
| 15–29 | 318/417 | 75.6(69.7–81.5) | 297/383 | 77.9(69.8–86.0) | 315/366 | 85.3(76.6–94.1) | 254/308 | 80.3(68.5–81.3) | 1184/1474 | 79.2(76.2–82.3) |
| Total | 711/1023 | 66.7(58.3–75.2) | 568/792 | 72(67.1–77.0) | 613/932 | 63.5(52.7–74.2) | 650/822 | 76.5(72.5–80.4) | 2542/3569 | 68.8(64.5–73.1) |
| **Age (years)** | | | | | | | | | | |
| 10–14 | 393/606 | 63.9(55.8–72.1) | 271/409 | 67.1(60.7–73.5) | 298/566 | 55.3(40.5–70.1) | 396/514 | 74.9(68.5–81.3) | 1358/2095 | 64.3(59.3–69.3) |
| 15–19 | 178/226 | 78.1(71.5–84.8) | 135/175 | 76.5(70.0–83.0) | 178/207 | 83.8(70.4–97.1) | 136/158 | 85.5(80.4–90.6) | 627/766 | 80.4(76.6–84.1) |
| 20–24 | 101/125 | 79(70.1–87.9) | 101/124 | 86(70.8–100) | 90/99 | 91.6(86.3–96.8) | 75/84 | 90.1(80.7–99.5) | 367/432 | 85.7(81.0–90.4) |
| 25–29 | 39/66 | 60.5(51.7–69.3) | 61/84 | 69.8(63.7–76.0) | 47/60 | 79.5(63.3–95.6) | 43/66 | 58.8(44.5–73.2) | 190/276 | 66.8(60.9–72.8) |
| Total | 711/1023 | 66.7 (58.3–75.2) | 568/792 | 72(67.1–77.0) | 613/932 | 63.5(52.7–74.2) | 650/822 | 76.5(72.5–80.4) | 2542/3569 | 68.8(64.5–73.1) |
| **Marital status** | | | | | | | | | | |
| Never married | 669/959 | 66.7(58.1–75.4) | 518/721 | 72.1(66.6–77.5) | 559/866 | 62.2(50.7–73.7) | 584/727 | 78.5(73.6–83.4) | 2330/3273 | 68.7(64.1–73.3) |
| Married | 38/60 | 64.2(54.5–74.0) | 47/66 | 72.3(54.1–90.6) | 52/63 | 88.2(74.2–100) | 42/56 | 63.4(30.5–96.3) | 179/245 | 72.1(62.8–81.5) |
| Divorced, Separated or Widowed | 1/1 | 100 (.—.) | 1/1 | 100 (.—.) | 0/0 | .(.—.) | 0/1 | .(.—.) | 2/3 | 42.4(0.0–100) |
| Total | 708/1020 | 66.6(58.3–75.0) | 566/788 | 72.1(67.2–77.0) | 611/929 | 63.5(52.6–74.3) | 626/784 | 77.6(73.5–81.7) | 2511/3521 | 68.9(64.5–73.2) |
| **Education level** | | | | | | | | | | |
| Primary and below | 516/771 | 64.6(56.2–73.1) | 386/571 | 68.2(61.5–74.8) | 438/732 | 59.2(46.2–72.1) | 504/659 | 74.3(69.2–79.4) | 1844/2733 | 65.7(61.1–70.4) |
| Secondary | 127/173 | 74.9(68.0–81.8) | 122/150 | 81(74.3–87.7) | 139/159 | 85.7(75.6–95.9) | 105/114 | 91.1(86.3–95.8) | 493/596 | 82(78.4–85.6) |
| Post-Secondary | 68/79 | 87.8 (76.6–99.1) | 60/71 | 83.6(37.0–100) | 36/41 | 92.5(78.0–100) | 41/49 | 79.6(59.4–99.8) | 205/240 | 85.7(74.4–97.1) |
| Total | 711/1023 | 66.7(58.3–75.2) | 568/792 | 72(67.1–77.0) | 613/932 | 63.5(52.7–74.2) | 650/822 | 76.5(72.5–80.4) | 2542/3569 | 68.8(64.5–73.1) |
| **Religion** | | | | | | | | | | |
| Christian | 693/1001 | 66.9(59.3–74.5) | 556/776 | 71.9(67.2–76.6) | 606/924 | 63.3(52.4–74.1) | 640/811 | 76.5(72.3–80.7) | 2495/3512 | 68.8(64.7–72.9) |
| Other | 18/22 | 59.2(0.0–100) | 12/16 | 78.9(32.0–100) | 7/8 | 93.5(55.7–100) | 10/11 | 73.4(3.0–100) | 47/57 | 69.1(39.0–99.1) |
| Total | 711/1023 | 66.7(58.3–75.2) | 568/792 | 72(67.1–77.0) | 613/932 | 63.5(52.7–74.2) | 650/822 | 76.5(72.5–80.4) | 2542/3569 | 68.8(64.5–73.1) |
| **Employment status** | | | | | | | | | | |
| Employed | 98/141 | 69.4(64.8–74.0) | 101/135 | 72(66.3–77.7) | 89/109 | 83.5(75.6–91.3) | 72/96 | 66.4(55.0–77.7) | 360/481 | 72(67.9–76.0) |
| Not employed | 613/882 | 66.3(56.7–75.9) | 467/657 | 72(66.6–77.5) | 524/823 | 61.6(49.8–73.4) | 578/726 | 77.6(73.3–81.8) | 2182/3088 | 68.3(63.5–73.2) |
| Total | 711/1023 | 66.7(58.3–75.2) | 568/792 | 72(67.1–77.0) | 613/932 | 63.5(52.7–74.2) | 650/822 | 76.5(72.5–80.4) | 2542/3569 | 68.8(64.5–73.1) |

(*Continued*)

**Table 4.** (Continued)

| Characteristic | Homa Bay | | Kisumu | | Migori | | Siaya | | Total | |
|---|---|---|---|---|---|---|---|---|---|---|
| | Unweighted | Weighted | Unweighted | Weighted | Unweighted | Weighted | Unweighted | Weighted | Unweighted | Weighted |
| | n/N | Prev. % (95% CI) | n/N | Prev. % (95% CI) | n/N | Prev. % | n/N | Prev. % | n/N | Prev. % |
| **Age** | | | | | | | | | | |
| **Ethnicity** | | | | | | | | | | |
| Luo | 690/1001 | 66.3(58.1–74.6) | 510/725 | 70.8(66.3–75.4) | 477/685 | 69.3(59.0–79.6) | 619/787 | 76.3(72.1–80.4) | 2296/3198 | 69.8(65.4–74.1) |
| Non-Luo | 21/22 | 97(89.9–100) | 58/67 | 84.5(67.4–100) | 136/247 | 48.1(26.0–70.2) | 31/35 | 85.3(66.5–100) | 246/371 | 58.9(40.2–77.7) |
| Total | 711/1023 | 66.7(58.3–75.2) | 568/792 | 72(67.1–77.0) | 613/932 | 63.5(52.7–74.2) | 650/822 | 76.5(72.5–80.4) | 2542/3569 | 68.8(64.5–73.1) |

who consented to physical verification among self-reported circumcised and uncircumcised respondents (95.8% vs. 95.0%; p-value = 0.39). Among 2,434 respondents who self-reported being circumcised, genital examination revealed that 99.5% (2,421/2,434) were circumcised, 0.2% (5/2,434) were partially circumcised and 0.3% (8/2,434) were uncircumcised. Similarly, of the 976 participants who reported they were uncircumcised and were examined, 98.6% (962/976) were confirmed to be uncircumcised, 0.4% (4/976) were found to be partially circumcised and 1.0% (10/1,027) were circumcised.

In total, nine respondents (0.3%) were found to be partially circumcised and four self-identified as uncircumcised while five self-identified as circumcised. Partially circumcised respondents were excluded from the analysis for agreement between self-reported and physically verified MC status. Overall agreement between self-reported and verified circumcision status was 99.2% (kappa agreement p <0.0001) with no significant differences across the four counties (not shown). Respondents who declined verification* or were found to be partially circumcised* were not included in the analysis for agreement between self-reported and physically verified MC status.

### 3.6 Predictors of circumcision

In bivariate analysis shown on Table 7, age, education, and county of residence were all significantly associated with verified circumcision status. Multivariate analysis revealed that men 15–19 years had 2.05 times higher odds of being circumcised compared to younger adolescents 10-14-year (95% CI [1.45, 2.89], p-value <0.001), while those aged 20–24 years had 2.33 times greater odds of being circumcised (95% CI [1.25, 3.65], p-value <0.006). Additional significant predictors of verified MC status included level of education, ethnicity and county of residence. Although ethnicity did not emerge as significantly associated with MC status in the bivariate analysis, it did when controlling for other covariates in the multivariate analysis.

**Table 5. Contribution of non-medical circumcision to the observed MC prevalence by age band and county.**

| Age | Siaya | | | Kisumu | | | Homabay | | | Migori | | |
|---|---|---|---|---|---|---|---|---|---|---|---|---|
| | MC Prev % | Medical (%) | Non-Medical (%) | MC Prev % | Medical (%) | Non-Medical (%) | MC Prev % | Medical (%) | Non-Medical (%) | MC Prev % | Medical (%) | Non-Medical (%) |
| 10-14yrs | 74.9 | 97.9 [95.1–99.1] | 2.1 [0.9–4.9] | 67.1 | 95.1 [90.6–97.5] | 4.9 [2.5–9.4] | 63.9 | 97.8 [94.8–99.1] | 2.2 [0.9–5.2] | 55.3 | 93.1 [83.1–97.4] | 6.9 [2.6–16.9] |
| 15-29yrs | 80.3 | 89.8 [68.2–97.3] | 10.2 [2.7–31.8] | 77.9 | 92.2 [83.7–96.4] | 7.8 [3.6–16.3] | 75.6 | 96.9 [93.5–98.6] | 3.1 [1.4–6.5] | 85.3 | 71.6 [56.6–83.0] | 28.4 [17.0–43.4 |

**Table 6. Self-reported circumcision status versus physically verified circumcision status among 3,569 boys and men in western Kenya, 2019.**

|  | Self-reported MC Status | | |
|---|---|---|---|
| Physical Verification outcome | Circumcised | Uncircumcised | Total |
| Declined verification* | 108 | 51 | 159 |
| Verified Circumcised | 2421 | 10 | 2431 |
| Verified Uncircumcised | 8 | 962 | 970 |
| Verified Partially Circumcised* | 5 | 4 | 9 |
| Total | 2542 (71.2%) | 1027(28.8%) | 3569 |

There was no difference between the overall MC prevalence based on self-report compared to physical verification in all counties.

## 4. Discussion

This population-based survey provided the latest MC prevalence data for boys and men aged 10–29 years in four non-circumcising counties with established VMMC programs in western Kenya. Consequently, the observed estimates are critical for setting new realistic targets for achieving the national goal of attaining and sustaining MC prevalence at 80% or more in the four counties.

**Table 7. Demographic predictors of verified male circumcision in four counties in western Kenya, 2019.**

| Covariate | Univariate analysis | | | | Multivariable analysis | | | |
|---|---|---|---|---|---|---|---|---|
|  | OR | 95% CI | P-value | Global P-value | OR | 95% CI | P-value | Global P-value |
| Age (n) | | | | | | | | |
| 10–14 years | Ref (1) | | | | | | | |
| 15–19 years | 2.53 | [1.82, 3.51] | <0.001 | <0.001 | 2.05 | [1.45, 2.89] | <0.001 | <0.001 |
| 20–24 years | 3.16 | [2.05, 4.88] | <0.001 | | 2.13 | [1.25, 3.65] | 0.006 | |
| 25–29 years | 1.05 | [0.74, 1.51] | 0.774 | | 0.68 | [0.44, 1.05] | 0.085 | |
| Marital Status (n) | | | | | | | | |
| Never Married | Ref (1) | | | | | | | |
| Married | 1.05 | [0.61, 1.81] | 0.858 | 0.663 | | | | |
| Separated/Divorced/Widowed | 0.33 | [0.03, 3.84] | 0.373 | | | | | |
| Education completed | | | | | | | | |
| Primary & below | Ref (1) | | | <0.001 | | | | <0.001 |
| Secondary | 2.72 | [1.95, 3.79] | <0.001 | | 1.77 | [1.31, 2.39] | <0.001 | |
| Post-secondary | 2.87 | [1.50, 5.51] | 0.002 | | 2.53 | [1.10, 5.80] | 0.029 | |
| Employment status | | | | | | | | |
| Employed | Ref (1) | | | 0.276 | | | | |
| Unemployed | 0.85 | [0.64, 1.14] | 0.276 | | | | | |
| Religion | | | | | | | | |
| Christian (ref) | Ref (1) | | | 0.979 | | | | |
| Other | 1.02 | [0.30, 3.42] | 0.979 | | | | | |
| Ethnicity | | | | | | | | |
| Luo | 1.72 | [0.87, 3.38] | 0.118 | 0.118 | 1.82 | [0.95, 3.49] | 0.07 | 0.07 |
| Non-Luo (ref) | Ref (1) | | | | | | | |
| County | | | | | | | | |
| Homa Bay (ref) | Ref (1) | | | 0.011 | | | | 0.023 |
| Kisumu | 1.31 | [0.92, 1.85] | 0.128 | | 1.19 | [0.84, 1.69] | 0.312 | |
| Migori | 0.85 | [0.50, 1.45] | 0.547 | | 0.97 | [0.58, 1.61] | 0.891 | |
| Siaya | 1.64 | [1.16, 2.33] | 0.006 | | 1.64 | [1.17, 2.31] | 0.005 | |

Using the observed MC prevalence from this survey as a reference, we concluded that the DMPPT2 model overestimated the MC prevalence in the four counties. The observed MC prevalence data across the two age bands (10–14 and 15–29 years) were 0.9–32.9 percentage points lower than the DMPPT2 modeled estimates for the same year. The survey results were also lower than the initial DMPPT2 estimates in 2016 by 5–10% [8]. This is despite additional VMMCs performed in the intervening period between the initial DMPPT2 modeling and this survey. The sustained high uptake of VMMC in age bands that had attained close to 100% MC prevalence by the DMPPT2 model three years earlier, supports lower MC prevalence as observed in this survey. By revealing that the DMPPT2 model overestimated the MC prevalence, this survey has resolved the inconsistency between sustained high uptake of VMMC and high MC prevalence from the 2016 modeling results published in 2018 [8]. Reasons for overestimation by the DMPPT2 model remain unclear but might include lack of precision in the age-specific baseline circumcision estimates and misreporting of client ages or the number circumcised in the program. The discrepancy between the MC prevalence estimates from the survey and DMPPT 2 was wider for 10-14-year age band (25.1–32.9%) than for 15-29-year-old clients (0.9–12.4%). This is a pointer to the possibility less accurate reporting of the ages for younger VMMC clients. Errors in the demographic model may also result in imprecision in the estimates; for example, if DMPPT2 is underestimating population growth. Furthermore, the DMPPT2 model assumes that the background rate of non-program circumcision remains constant over time. Errors in adjusting the model inputs to account for any changes in the contribution of traditional circumcision to the total numbers can result in overestimation of prevalence.

The gaps in MC prevalence revealed by this survey across the VMMC priority counties provide a good basis for refocusing program geographically based on unmet need. Although the VMMC program target of 80% MC prevalence was reached or exceeded among 15–29 year old men in Siaya (80.3%) and Migori counties (85.3%), and nearly achieved in Kisumu (77.9%) and Homa Bay (74.6%), the prevalence among young adolescents aged 10–14-years was lower in all counties surveyed. Therefore, Kenya requires sustained investment in VMMC to address need for men aged 15 years or more years and to maintain services for progressively larger annual cohorts young adolescents who become eligible for VMMC annually due to the youth bulge demographic phenomenon [16, 17].

Although this survey showed MC prevalence of 55.3–74.9% among young adolescents aged 10–14 years, this is bound to decline over time if the country refocuses VMMC services towards older males as recommended by WHO guidelines of 2020. Both WHO and PEPFAR have deemphasized VMMC services for boys below 15 years due to concerns over safety and challenges of consenting [18, 19]. This policy shift, coupled with the youth bulge demographic phenomenon, likely lead to a progressive increase in the absolute number and proportion of boys turning 15 years before being circumcised. Consequently, Kenya will need to set progressively higher VMMC targets for the 15-19-year-old adolescents to minimize the dilution effect of the increasing number of uncircumcised boys graduating into this age band.

As expected, the observed MC prevalence was higher in the 15–29 than 10-14-year age group in all counties with the prevalence gap ranging from 5–30 percentage points. The prevalence gap between the two age groups was largest in Migori at 30.0% compared to 5.4–11.7% in the remaining three counties. A likely explanation for the steeper MC prevalence increase in Migori after 14 years is the preference by culturally circumcising residents for men to be circumcised between 15 and 18 years. Unlike Kisumu, Siaya and Migori, three out of the nine sub-counties in Migori are inhabited partially by ethnic group (the Kuria) who prefer that men be circumcised from 15 to 18 years. The observed contribution of non-medical circumcision to the overall MC prevalence for men in the 15-29-year age group ranged from 3.1% in Homa

bay to 28.4% in Migori. Similarly preliminary results of 2018 Kenya population-based impact assessment (KENPHIA 2018) [4] also showed that the contribution of non-medical circumcision to the overall MC prevalence was highest in Migori(24.4%) compared to 2.7–6.6% in the remaining three counties. These observations underscore the importance of accounting for the contribution of non-medical circumcision and preferred age of circumcision for different subgroups when assessing the effect of VMMC on the population MC prevalence in counties occupied by both circumcising and non-circumcising subgroups.

A secondary objective of this survey was to assess the level of agreement between self-reported and physically verified circumcision status. There was high participation in physical verification of MC status, and high agreement of 99.2% between self-reported and physically verified MC status. An earlier study conducted in the same counties in 2014–2015 found comparable agreement of 98.6% among 24–39 year-old men [9]. Compared to this study, in which the overall response rate was 84.6% with 95.5% of the respondents consenting to physical verification, the earlier survey had a low participation rate (58.3%), but comparable uptake of physical verification (97.8%). Our findings support the use of self-reported circumcision status as a reliable source of data for estimating the population prevalence of MC.

On physical examination, we found nine respondents who were partially circumcised, but it was uncertain if these were naturally short or incompletely removed foreskins. Four partially circumcised respondents who self-identified as uncircumcised and were presumed to be cases of naturally short foreskin while five who reported that they were circumcised were presumed to have had incomplete removal of the foreskin during circumcision. Though rare, incomplete removal of the foreskin is a significant adverse event [20] because it can theoretically lower the HIV prevention efficacy of VMMC [21]. Programs should guard against incomplete excision of the foreskin because it can erode public confidence in the VMMC program besides undermining its HIV prevention benefits. Strategies for preventing insufficient removal of the foreskin include consistent supervision and support for health workers through refresher training. Use of male circumcision devices such as ShangRing may also standardize the amount of foreskin removed and minimize the risk of insufficient skin removal [22]. Partial circumcisions were randomly distributed across the counties regardless of the prevalence of non-medical circumcision and with no clustering among non-medically circumcised respondents compared to those circumcised medically.

Partial circumcision also has practical implications for surveys that involve physical verification of MC status because of the need to distinguish between the incomplete removal of the foreskin, which is preventable through proper circumcision techniques, and naturally short foreskin. Such surveys should include questions on the date and place of circumcision procedure to permit review of patient records, if needed, to ascertain any history of past circumcision.

In the multivariable model, age, level of education, ethnicity and county of residence were significant independent predictors of circumcision status. Specifically, having a higher level of education, being aged 15–24 years and being of Luo ethnic community were associated with being circumcised. These findings are consistent with the results of a similar survey conducted in the same counties in 2014–2015 among men aged 25–29 years [23]. In that study, higher education and having employment were associated with being circumcised, and having ever been married (currently married, divorced, separated, or widowed) was associated with being uncircumcised. Our results support the need for tailored education on benefits of VMMC and mobilization targeting men with lower education and other demographic subgroups with lower likelihood of being circumcised.

This study had some limitations; 1). There were only two response options for self-reported circumcision status (circumcised or uncircumcised) but physical verification included a third

outcome of partial circumcision. Fortunately, partial circumcisions were few (9 out of 4,010) and were excluded from the analysis of agreement between self-reported and physically veri-fied MC status without affecting the overall results. The observed MC prevalence data in this survey conducted in 2019 were compared with DMPPT2 modelled estimates of 2016 and found to be lower. The margin of MC prevalence overestimation in DMPPT2 would probably be larger had we compared the results with DMPPT2 modeled estimates for 2019. Necessary adjustments for migration and replacement for DMMPT2 inputs beyond 2016 were not com-pleted in time for consideration in this analysis. 3). It is possible that some clients who had incomplete removal of the foreskin failed to disclose their history of previous circumcision due to embarrassment. 4). The study was primarily conducted among the non-circumcising Luo ethnic group where MC has been promoted as a medical intervention for HIV prevention. The results of agreement between self-reported and physically verified circumcision status may not be generalizable to settings where VMMC program is not fully embraced because low social desirability and social disapproval of MC may discourage respondents from disclosing their correct status.

## 5. Conclusion

Using the observed population prevalence of MC from this survey as a reference, we have dem-onstrated that the initial DMPPT2 modeling performed in Kenya in 2016 and published in 2018 provided inflated estimates of MC prevalence especially for men aged 15–29 years. Regardless of the basis for the previous overestimates, this survey has provided up to date MC prevalence data which form a good reference for setting realistic VMMC program targets and re-calibrating inputs into DMPPT2. Similar population-based MC prevalence surveys con-ducted periodically, especially for mature programs, can help reconcile inconsistencies between VMMC program uptake data and modeled MC prevalence estimates which are based on the number of procedures reported in the program annually.

## Supporting information

**S1 File. Appendix-7 questionnaire English.**
(PDF)

**S2 File. Appendix 7_questionnaire_Kiswahili.**
(PDF)

**S3 File. Appendix 7_questionnaire_Dholuo.**
(PDF)

**S1 Table. 2019 Kenya MC survey data dictionary.**
(PDF)

**S2 Table. 2019 Kenya MC survey dataset.**
(CSV)

## Acknowledgments

This survey was jointly implemented by Jhpiego, Kenya's Ministry of Health and the national VMMC technical working group. The County Governments of Siaya, Kisumu, Homa bay and Migori and their respective implementing partners, namely, CHS, UCSF, EGPAF and UMB. The Kenya National Bureau of statistics (KNBS) provided technical assistance in sampling and data weighting. We are grateful to all these agencies, the survey personnel and the residents of

Siaya, Kisumu Homa bay and Migori Counties for their diverse contributions towards the success of this survey.

**Disclaimer:** The findings and conclusions in this report are those of the authors and do not necessarily represent the official position of the funding agencies.

## Author Contributions

**Conceptualization:** Elijah Odoyo-June, Stephanie Davis, Nandi Owuor, Catey Laube, Kawango Agot.

**Data curation:** Jonesmus Wambua, Paul Musingila.

**Formal analysis:** Jonesmus Wambua, Paul Musingila, Peter W. Young.

**Investigation:** Nandi Owuor.

**Methodology:** Elijah Odoyo-June, Stephanie Davis.

**Project administration:** Nandi Owuor, Catey Laube, Zebedee Mwandi.

**Software:** Jonesmus Wambua.

**Supervision:** Elijah Odoyo-June, Nandi Owuor, Catey Laube, Jonesmus Wambua, Appolonia Aoko, Kawango Agot, Zebedee Mwandi.

**Validation:** Paul Musingila.

**Writing – original draft:** Elijah Odoyo-June.

**Writing – review & editing:** Elijah Odoyo-June, Stephanie Davis, Nandi Owuor, Catey Laube, Peter W. Young, Appolonia Aoko, Kawango Agot, Rachael Joseph, Zebedee Mwandi, Vincent Ojiambo, Todd Lucas, Carlos Toledo, Ambrose Wanyonyi.

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
