## [Decision Letter · Decision Letter 0]

16 Feb 2021

PONE-D-20-37607

Prevalence of male circumcision in four culturally non-circumcising counties in western Kenya after 10 years of program implementation from 2008 to 2019

PLOS ONE

Dear Dr. Odoyo-June,

Thank you for submitting your manuscript to PLOS ONE. After careful consideration, we feel that it has merit but does not fully meet PLOS ONE’s publication criteria as it currently stands. Therefore, we invite you to submit a revised version of the manuscript that addresses the points raised during the review process.

Please be sure to

1) carefully consider the comments about the interplay between the observed data and the modelling estimates in your reporting. It is not clear whether this plays a large or a small role in the work and the conclusions.

2) Reduce commentary and speculation about partial circumcision as his was almost wholly unobserved.

We look forward to receiving your revised manuscript.

Kind regards,

Deborah Donnell, Ph. D.

Academic Editor

PLOS ONE

Journal Requirements:

2. Please include additional information regarding the survey or questionnaire used in the study and ensure that you have provided sufficient details that others could replicate the analyses. For instance, if you developed a questionnaire as part of this study and it is not under a copyright more restrictive than CC-BY, please include a copy, in both the original language and English, as Supporting Information, or include a citation if it has been published previously.

3. Please include further description of the questionnaire development process in your methods section.

4. In the Methods, please discuss whether and how the questionnaire was validated and/or pre-tested. If these did not occur, please provide the rationale for not doing so.

5. In statistical methods, please refer to any post-hoc corrections to correct for multiple comparisons during your statistical analyses. If these were not performed please justify the reasons. Please refer to our statistical reporting guidelines for assistance (https://journals.plos.org/plosone/s/submission-guidelines.#loc-statistical-reporting).

"This survey was funded by the President’s Emergency Plan for AIDS Relief (PEPFAR) through the U.S.

361 Centers for Disease Control and Prevention (CDC) under the terms of Cooperative Agreement #

362 GH001469 and was jointly implemented by Jhpiego and Kenya’s Ministry of Health."

"The author(s) received no specific funding for this work.

7. We note you have included a table to which you do not refer in the text of your manuscript. Please ensure that you refer to Table 5 in your text; if accepted, production will need this reference to link the reader to the Table.

Additional Editor Comments:

Overall, this is a nicely summarized piece of work providing important data on current prevalence of male circumcision, that is an important reality check for mathematical modelling and better understanding of demand for services. The authors are to be congratulated on high quality of study design and execution, and the appropriate statistical analysis for the design.

Overall

1. There are several places where the text is not clear about the role and predictions of the DMPPT2 modelling estimates. In the back ground and introduction much greater clarity about the mathematical modelling issues is needed to better understand the context for the work.

• Second sentence of abstract introduction needs revision – this reviewer could not understand

• No context for the “lower population-based MC prevalence” is given in the abstract – I think the context of overly optimistic model projections is needed

• Introduction: More precision in the language is needed to make it clear when estimates are “model estimates” and when estimates are from observed data. For example mention of “VMMC program saturation” – it is not clear whether this is estimated saturation or field (observed) saturation.

2. Another area that needs clarity is whether “MC” combines both traditional and VMMC. It appears the MC reported is almost all VMMC, but this needs to be discussed.

Methods: “none selection” should be “non-selection”

Statistical: The sampling design and the sample size were very similar in different regions. It seems surprising that the 95%CI for 10-14 yo in Migoro was ~ +/- 15%, but for Siaya was +/- 8%. Typically the SE would not change so dramatically for prevalence in the range ~55% to ~75%.

Tables:

• In Table 1 and Table 2 row% should be replaced by col%, as there is little interest in comparison between regions.

• In Table 1: Please change to % interviewed within each subgroup; similarly the last two rows could be entered as % fof the two rows above rather than as separate rows.

Discussion:

• Several instances of problems with references and words that were left in from editing “(insert coverage…)”

• Line 274-278: This is the first time details are provided about the modelled estimates. It would be helpful to include this in the introduction. The statement that the observed estimates were lower than the modelled estimates is incorrect – because they are within the range provided for the 10-14 yos, simply on the lower end. The range is, of course very wide (so wide as to be not very useful?).

• Line 287ff. This reviewer understood that traditional MC in the South African is typically partial, and there were extremely few partial circumcisions reported. It would be helpful in the commentary in the discussion about the potential role of traditional circumcision in Migori if this could be clarified, as it seems somewhat in conflict with the explanation offered.

Reviewers' comments:

Reviewer's Responses to Questions

**Comments to the Author**

1. Is the manuscript technically sound, and do the data support the conclusions?

Reviewer #1: Yes

2. Has the statistical analysis been performed appropriately and rigorously? 

Reviewer #1: Yes

3. Have the authors made all data underlying the findings in their manuscript fully available?

Reviewer #1: Yes

4. Is the manuscript presented in an intelligible fashion and written in standard English?

Reviewer #1: Yes

5. Review Comments to the Author

Reviewer #1: Odoyo-June et al. describe the results of a survey of circumcision prevalence among boys and men in four counties of Kenya. The manuscript is well-written. I have only minor suggestions for how to improve the clarity and flow of the manuscript.

Title

Comparing the title to the abstract, it was difficult to tell whether the purpose of the study was to ascertain VMMC prevalence in four counties of Kenya (title) or to validate the DMPPT2 (abstract). It is fine for there to be a secondary purpose but best to be consistent between title and abstract as to the main purpose of the study.

Abstract

The abstract is inconsistent as described above. Introduction implies that the purpose is to validate DMPPT2, but Results only discuss VMMC prevalence in the four counties and Conclusion does not help readers interpret the results in terms of whether DMPPT2 was validated.

The abstract should be revised to ensure it makes sense as a stand-alone document. I recommend reading it with fresh eyes and revising for clarity. Here are some issues that stood out to me as a new reader:

"80% saturation" -- what does this mean? Why is 80% a point of saturation?

"Estimates from the DMPPT2 included unlikely MC coverage approaching 100% without commensurate decline in VMMC uptake." -- I could not understand the meaning of this sentence

From reading conclusions I could not tell whether the survey results did or did not agree with the DMPPT2 estimates. Clearly state this and discuss possible reasons why or why not.

Introduction

"The program achieved 92% of its target" -- please clarify what the target is.

Methods

line 109 "using probability proportional to size sampling" is unclear. Is this probability in proportion to the total population, males age 10+, land area? Please specify and, if possible, provide a reference to the complete method.

line 111 "All households in the sampled EAs were listed to help update NASSEP IV" is unclear. Did a staff member manually enumerate each HH? Was this list used directly or merged with the NASSEP list? If merged, how was it deduplicated?

The survey weights description was excellent and very clear except for this part on Lines 173-174: "Finally, the weights were adjusted to ensure consistency with the projected population figures." It was not clear to me what this meant methodologically.

The long sentence starting on Line 176 is missing a subject, and could generally be re-written for readability/clarity:

County population MC prevalence (both verified and self-reported) was calculated by multiplying the value of each participant’s survey response by the corresponding nonresponse-adjusted [***MISSING SUBJECT***] summing up the products across all units (clusters and strata) and finally dividing by the sum of all weights.

Results

Given the claim that the survey resolved discrepancies with DMPPT, Results should also include results from DMPPT and whether they agree with the data given sampling uncertainties in the survey. Some of the DMPPT results are included in Discussion instead which makes for an awkward flow.

Table 3 is the "meat" of the paper and it is very large and difficult to navigate. A figure and a map would be welcome. Maybe a map of counties colored by prevalence, overlayed with dots representing EAs also colored by prevalence.

Discussion

As best I understand, PEPFAR no longer recommends circumcision for ages 10-14 due to higher risk of adverse events. The paper should address this. For example, the observation that 15-19yo boys are 2x more likely to become circumcised than 10-14yo boys should be discussed in the context of the recent guidance.

The Discussion section has multiple instances of "(insert coverage for these counties)" and "Error! Bookmark not defined."

DMPPT results should be in Results instead of Discussion.

A large section of the Discussion focuses on incomplete circumicision. This part of the paper is both confusing and concerning. It speaks of nine individuals with short but not entirely removed foreskins, says four of them stated they were uncircumcised, and for some reason deduces that four of them were partially circumcised. Why four and not five?

It then discusses how these might be VMMC adverse events and may erode confidence in VMMC programs and discusses the potential benefits of ShangRing in avoiding partial removal. Although the authors may be trying to speak in hypotheticals, this really feels like jumping to conclusions. It may be best to omit this paragraph in order to not cast doubt on the Kenyan VMMC program, since these results may be spurious. Follow-up with this handful of partially circumcised participants seems warranted in order to verify partial circulcision, interview respondents regarding their history of circumcision procedures, and alert the VMMC program if there are quality issues.

Finally, the authors claim that "Partial circumcisions have practical implications for surveys." However, it seems that the number of partially circumcised individuals in this study was extremely small, so much so that there may not be significant implications for surveys from this finding. It seems important to point that out. To address this formally, the authors could perform sensivitity analyses to determine (1) if the partially circumcised participants would have any meaningful impact on the survey results, and (2) if partial circumcision would meaningfully reduce the coverage of VMMC. If it is very rare and not a driver of survey bias or coverage, then it does not seem appropriate to single it out as a main topic of Discussion.

6. PLOS authors have the option to publish the peer review history of their article (what does this mean?). If published, this will include your full peer review and any attached files.

Reviewer #1: **Yes: **Anna Bershteyn

---

## [Author Response · Author response to Decision Letter 0]

10 Mar 2021

All review comments have been addressed.

Summary of responses is uploaded under filename Response to Reviewers.

The manuscript has been edited in response to review comments. A tracked and clean version uploaded.

---

## [Decision Letter · Decision Letter 1]

27 Apr 2021

PONE-D-20-37607R1

Prevalence of male circumcision in four culturally non-circumcising counties in western Kenya after 10 years of program implementation from 2008 to 2019

PLOS ONE

Dear Dr. Odoyo-June,

Thank you for submitting your manuscript to PLOS ONE. After careful consideration, we feel that it has merit but does not fully meet PLOS ONE’s publication criteria as it currently stands. Therefore, we invite you to submit a revised version of the manuscript that addresses the points raised during the review process.

Ed:  Please accommodate the remaining minor reviewers suggestions, and I will be happy to accept your MS for publication.

We look forward to receiving your revised manuscript.

Kind regards,

D W Cameron, MD

Academic Editor

PLOS ONE

Journal Requirements:

Additional Editor Comments (if provided):

Please accommodate the remaining reviewers suggestions, and I will be happy to accept your MS for publication.

Reviewers' comments:

Reviewer's Responses to Questions

**Comments to the Author**

1. If the authors have adequately addressed your comments raised in a previous round of review and you feel that this manuscript is now acceptable for publication, you may indicate that here to bypass the “Comments to the Author” section, enter your conflict of interest statement in the “Confidential to Editor” section, and submit your "Accept" recommendation.

Reviewer #1: All comments have been addressed

Reviewer #2: All comments have been addressed

2. Is the manuscript technically sound, and do the data support the conclusions?

Reviewer #1: Yes

Reviewer #2: Yes

3. Has the statistical analysis been performed appropriately and rigorously? 

Reviewer #1: Yes

Reviewer #2: Yes

4. Have the authors made all data underlying the findings in their manuscript fully available?

Reviewer #1: Yes

Reviewer #2: Yes

5. Is the manuscript presented in an intelligible fashion and written in standard English?

Reviewer #1: Yes

Reviewer #2: Yes

6. Review Comments to the Author

Reviewer #1: The paper is much improved, addressing all of my concerns. I have just one minor remaining concern.

In the re-written manuscript, the DMPPT estimates are stated in the text as % over-estimation relative to the survey for each of the Counties in the study. The DMPPT estimates themselves are not listed anywhere. For ease of reference, the VMMC results table should include a column with the DMPPT estimate and can then have a column with the % over-estimation relative to the survey. Otherwise, it is too difficult for a reader to pull these figures out of the text without having a table to refer to.

Reviewer #2: This is a revision of an analysis investigating the prevalence of male circumcision and associated characteristics in four counties in Kenya. This manuscript has been previously reviewed and it appears in the response to reviewers that the authors have addressed the concerns with the changes reflected in the edited manuscript. I recommend reviewing the tables and suppressing cells less than 5 to protect privacy.

7. PLOS authors have the option to publish the peer review history of their article (what does this mean?). If published, this will include your full peer review and any attached files.

Reviewer #1: **Yes: **Anna Bershteyn

Reviewer #2: No

---

## [Author Response · Author response to Decision Letter 1]

11 Jun 2021

Response to comments by Reviewer No 1. 

We have included a table that compares the 2019 DMPPT 2 modelled MC prevalence estimates with the population survey results for the same year (Table 3 on page 16; Line 256-258).

Response to comments by Reviewer No 2

Tables 2 and 3 have marital status variable which has three categories; “Divorced, separated or Widowed” which have cell values less than 5. Kindly note that this category cannot be merged with either of the two remaining categories because Never married and Married are standalone. Therefore, this category (Divorced, separated or Widowed) of marital status with cell values less than 5 can only remain as is and only used in descriptive and weighting cases only.

---

## [Decision Letter · Decision Letter 2]

21 Jun 2021

Prevalence of male circumcision in four culturally non-circumcising counties in western Kenya after 10 years of program implementation from 2008 to 2019

PONE-D-20-37607R2

Dear Dr. Odoyo-June,

We’re pleased to inform you that your manuscript has been judged scientifically suitable for publication and will be formally accepted for publication once it meets all outstanding technical requirements.

Kind regards,

D W Cameron, MD

Academic Editor

PLOS ONE

Additional Editor Comments (optional):

Good work.

Reviewers' comments:

Reviewer's Responses to Questions

**Comments to the Author**

1. If the authors have adequately addressed your comments raised in a previous round of review and you feel that this manuscript is now acceptable for publication, you may indicate that here to bypass the “Comments to the Author” section, enter your conflict of interest statement in the “Confidential to Editor” section, and submit your "Accept" recommendation.

Reviewer #1: All comments have been addressed

Reviewer #2: All comments have been addressed

2. Is the manuscript technically sound, and do the data support the conclusions?

Reviewer #1: Yes

Reviewer #2: Yes

3. Has the statistical analysis been performed appropriately and rigorously? 

Reviewer #1: Yes

Reviewer #2: Yes

4. Have the authors made all data underlying the findings in their manuscript fully available?

Reviewer #1: Yes

Reviewer #2: Yes

5. Is the manuscript presented in an intelligible fashion and written in standard English?

Reviewer #1: Yes

Reviewer #2: Yes

6. Review Comments to the Author

Reviewer #1: All comments have been addressed and the paper appears ready for publication. Table 3 makes it much easier to compare the estimates to the model projections. Thank you for adding it.

Reviewer #2: Thank you for the opportunity to review this manuscript. All my comments have been addressed. It is both well designed and clearly written.

7. PLOS authors have the option to publish the peer review history of their article (what does this mean?). If published, this will include your full peer review and any attached files.

Reviewer #1: **Yes: **Anna Bershteyn

Reviewer #2: No

---

## [Editor Report · Acceptance letter]

24 Jun 2021

PONE-D-20-37607R2 

Prevalence of male circumcision in four culturally non-circumcising counties in western Kenya after 10 years of program implementation from 2008 to 2019 

Dear Dr. Odoyo-June:

I'm pleased to inform you that your manuscript has been deemed suitable for publication in PLOS ONE. Congratulations! Your manuscript is now with our production department. 

Kind regards, 

on behalf of

Professor D W Cameron 

Academic Editor

PLOS ONE